Investigating the potential impact of 1.5, 2 and 3 °C global warming levels on crop suitability and planting season over West Africa

Egbebiyi Temitope Samuel EGBTEM001@myuct.ac.za 1
Crespo Olivier 1
Lennard Christopher 1
Zaroug Modathir 1 5 6
Nikulin Grigory 2
Harris Ian 3
Price Jeff 4
Forstenhäusler Nicole 4
Warren Rachel 4
1 Climate System Analysis Group (CSAG), Department of Environmental and Geographical Science, University of Cape Town , Cape Town , Western Cape , South Africa
2 Swedish Meteorological and Hydrological Institute , Norrköping , Sweden
3 Climate Research Unit, School of Environmental Sciences, University of East Anglia , Norwich , United Kingdom
4 Tyndall Centre for Climate Change Research, School of Environmental Sciences, University of East Anglia , Norwich , United Kingdom
5 African Climate and Development Initiative (ACDI), University of Cape Town , South Africa
6 Nile Basin Initiative Secretariat , Entebbe , Uganda
Raga Graciela
Electronic publication date: 2020 May 5
Publication date: 2020
Volume: 8
Electronic Location ID: e8851
Received 2019 Sep 2; Accepted 2020 Mar 4
Copyright: ©2020 Egbebiyi et al.
Copyright year: 2020
Copyright holder: Egbebiyi et al.
License: This is an open access article distributed under the terms of the Creative Commons Attribution License, which permits unrestricted use, distribution, reproduction and adaptation in any medium and for any purpose provided that it is properly attributed. For attribution, the original author(s), title, publication source (PeerJ) and either DOI or URL of the article must be cited.
License URL: https://creativecommons.org/licenses/by/4.0/

Keywords: Global warming levels, 1.5, 2.0 & 3.0 °C; Crop suitability; Planting season; Ecocrop; CORDEX; West Africa; Food security

Funding: National Research Foundation (NRF, South Africa) Tyndall Centre for Climate Change and Climatic Research Unit (CRU) under the Newton PhD Partnering Scheme funded by Research Councils United Kingdom (RCUK) Alliance Centre for Climate and Earth Systems Science (ACCESS, South Africa) African Climate and Development Initiative (ACDI) This research was supported by funding from the National Research Foundation (NRF, South Africa). the Tyndall Centre for Climate Change and Climatic Research Unit (CRU) under the Newton PhD Partnering Scheme funded by Research Councils United Kingdom (RCUK), the Alliance Centre for Climate and Earth Systems Science (ACCESS, South Africa), the African Climate and Development Initiative (ACDI), the JW Jagger Centenary Scholarship and the Siri Johnson scholarship, University of Cape Town, South Africa. The funders had no role in study design, data collection and analysis, decision to publish, or preparation of the manuscript.

==============================
West African rainfed agriculture is highly vulnerable to climate variability and change. Global warming is projected to result in higher regional warming and have a strong impact on agriculture. This study specifically examines the impact of global warming levels (GWLs) of 1.5°, 2° and 3 °C relative to 1971–2000 on crop suitability over West Africa. We used 10 Coupled Model Intercomparison Project Phase5 Global Climate Models (CMIP5 GCMs) downscaled by Coordinated Regional Downscaling Experiment (CORDEX) Rossby Centre’s regional Atmospheric model version 4, RCA4, to drive Ecocrop, a crop suitability model, for pearl millet, cassava, groundnut, cowpea, maize and plantain. The results show Ecocrop simulated crop suitability spatial representation with higher suitability, observed to the south of latitude 14°N and lower suitability to its north for 1971–2000 for all crops except for plantain (12°N). The model also simulates the best three planting months within the growing season from September-August over the past climate. Projected changes in crop suitability under the three GWLs 1.5–3.0 °C suggest a spatial suitability expansion for legume and cereal crops, notably in the central southern Sahel zone; root and tuber and plantain in the central Guinea-Savanna zone. In contrast, projected decreases in the crop suitability index value are predicted to the south of 14°N for cereals, root and tuber crops; nevertheless, the areas remain suitable for the crops. A delay of between 1-3 months is projected over the region during the planting month under the three GWLs for legumes, pearl millet and plantain. A two month delay in planting is projected in the south, notably over the Guinea and central Savanna zone with earlier planting of about three months in the Savanna-Sahel zones. The effect of GWL2.0 and GWL3.0 warming in comparison to GWL1.5 °C are more dramatic on cereals and root and tuber crops, especially cassava. All the projected changes in simulated crop suitability in response to climatic variables are statistically significant at 99% confidence level. There is also an increasing trend in the projected crop suitability change across the three warming except for cowpea. This study has implications for improving the resilience of crop production to climate changes, and more broadly, to food security in West Africa.

Introduction

Rainfed agriculture is crucial to the economy and livelihood of the inhabitants of West Africa (Omotosho & Abiodun, 2007; Roudier et al., 2011; Diasso & Abiodun, 2017). The agricultural sector employs more than 65% of the active labour force in the region, with the majority practising subsistence rain-fed farming (Benhin, 2008; Schlenker & Lobell, 2010; Roudier et al., 2011). The sector is also responsible for 75% of Sub-Saharan Africa (SSA) domestic trade (McCarthy et al., 2001; World Bank, 2013) and contributes significantly to the economy via a Gross Domestic Product (GDP) of up to 20% (World Bank Report, 2009; Schlenker & Lobell, 2010; Roudier et al., 2011). Like other regions of Africa, West Africa has suffered devastating effects of change to from climate change impacts via rainfall variability and droughts due to global warming in the last few decades (Sarr, 2012; Diasso & Abiodun, 2017). Thus, further warming over the region could worsen the current climatic stress on agricultural production, the main source of livelihood of the inhabitants.

Global mean surface temperature has increased by approximately 1 °C above pre-industrial levels and is likely to rise to 1.3–4.8 °C by 2081–2100 (IPCC, 2013; IPCC, 2018). It is projected that the average temperature increase will be more intense in Africa than the rest of the globe (Solomon et al., 2007; Sarr, 2012). For the West African region, observed temperatures have increased between 0.2 and 0.8 since the end of 1970s a trend which is stronger with minimum temperature and faster than global warming (Sarr, 2012). This trend is significant and much higher than the global warming trend (Sarr, 2012). Furthermore, other studies have revealed that an increase in global warming will result in the deviation of the mean temperature from the historical variability leading to a new climate regime over the continent, particularly West Africa (Hawkins & Sutton, 2012; Mora et al., 2013). This deviation from historical variability was reaffirmed by a study by Mora et al. (2013) which went a step further to show that the mean temperature over West Africa will move outside the bounds of historical variability in the next two decades earlier before the global mean temperature thus making the region a hotspot from the impact of global warming. In addition, a World Bank report revealed that 2–4 °C of warming poses a threat to agriculture and food security in sub-Saharan Africa (World Bank, 2013). This projected warming is expected to affect the agricultural sector by a reduction of up to 50% in crop yield and 90% in revenue by the end of the century (IPCC, 2013; World Bank, 2013). Thus, knowing what the projected impacts of 1.5–3 °C warming are above pre-industrial level are on crop growth suitability over West Africa is of great importance.

The United Nations Framework Convention on Climate Change (UNFCCC) Paris Agreement aims to limit global average temperature rise to ‘well below 2 °C above pre-industrial levels’ and to pursue efforts to limit it to 1.5 °C (UNFCCC, 2015) and the scientific community has since explored methods to help achieve this goal (IPCC, 2018). The signing of this Agreement was hinged on the submission of the Intended Nationally Determined Contributions (INDC) documents by each member country stating their plan for addressing climate change beyond 2020 by limiting the global mean temperature below 2 °C (Rogelj et al., 2016). The INDC document, now Nationally Determined Contribution (NDC), after the Paris Agreement, addresses issues such as avoiding, adapting to, coping with climate change challenge (Rogelj et al., 2016; Hof et al., 2017). Although, the aim of the Paris Agreement as expressed in the NDCs is to limit global warming to 1.5 °C and well below 2.0 °C, they are inadequate to do so. Furthermore, the trends in CO2 emissions indicates an urgent decline in global emission is crucial to the possibility of reducing warming below 2 °C (Luderer et al., 2013; Rogelj et al., 2013) while about two-third of the available resource to keep warming to below 2 °C have already been emitted (Meinshausen et al., 2009; IPCC, 2014). Thus, the NDCs would likely mean that global temperatures may increase by 3 °C or more; below the necessary emission reduction consistent with 2 °C and 1.5 °C climate target (Rogelj et al., 2016; Hof et al., 2017). The impact of 1.5–2 °C warming is projected to be more pronounced in regions with low adaptive capacity and high exposure, such as West Africa (Mitchell et al., 2016; Rogelj et al., 2016; Schleussner et al., 2016). For example, in a tropical region like West Africa, holding the temperature increase below 1.5 °C has a positive impact of limiting local yield reduction in wheat and maize (Schleussner et al., 2016). However, no previous studies have examined how the impact of these three different global warming levels, 1.5, 2.0 & 3.0 °C suggested by the Intergovernmental Panel on Climate Change (IPCC) tied to policy aspirations and goals, will affect crop mean suitability growth and planting months within a growing season in this region. In addition, no study has examined how the impact of the inability to meet the NDC plan and the potential of reaching a projected temperature increase up to 3 °C (Mora et al., 2013; Rogelj et al., 2016) will affect crop suitability and agriculture over West Africa.

Given the global level significance of this threshold, and its particularly high exposure, the aim of this study is to examine the potential implications of the global warming levels (GWLs) 1.5, 2.0 & 3.0 °C on crop suitability and month of planting in West Africa. Crop suitability in this study is described as the appropriateness of an area of land based on the growing threshold of a crop in relation to climatic condition, minimum and mean monthly temperature and total monthly rainfall (FAO, 1976; Singh et al., 2018; Egbebiyi et al., 2019). It refers to the spatial appropriateness distribution of the land area based on the growing climatic suitability threshold of a crop over given time period.

We examined how the differences between GWL1.5, 2.0 & 3.0 °C could influence crop growth suitability over West Africa to assess the benefit of limiting global warming. This information is important for developing timely adaptation strategies to improve crop yield and food security in the region. ‘Data and Methods’ describes the study area, the climate variables and crop data; it also gives an overview of the crop suitability model, Ecocrop used for the study. ‘Results’ describes Ecocrop suitability results for the historical climate, GWLs1.5, 2.0 & 3.0 °C and the difference between them. In ‘Discussion’, the results from the study are discussed in relation to improving food security and adaptation strategies over West Africa. The concluding remarks are given in ‘Summary and Conclusion’.

Data and Methods

Study domain

The study area is West Africa, which has rainfed agriculture as its mainstay economy. It ranges from latitude 2–20°N and 20°W to 20°E (Fig. 1). The region is divided into three agro-ecological zones, namely, Guinea (4–8°N), Savanna (8–12°N) and the Sahel (12–20°N) (FAO, 2005; Bourn, 2013). The temperature gradient over the region increases to the north, while precipitation increases to the south in the region. The West Africa Monsoon (WAM) is the major system influencing the rainfall pattern in West Africa (Omotosho & Abiodun, 2007; Nicholson, 2013). The region provides a large amount of agricultural resources. However, due to its variability in rainfall patterns and low adaptive capacity, this region faces substantial risk from climate change (Williams, Crespo & Abu, 2018).

Figure 1 The West African Agro-Ecological Zones designated as Guinea, Savanna and Sahel respectively. Map credit: Livestock in a Changing Landscape (2010). Copyright ©2010 Scientific Committee on Problems of the Environment (SCOPE).

Reproduced by permission of Island Press, Washington, D.C.

Different parts of West Africa cultivate and grows different crops which contributes to the economy of the region. Major crops grown in the region include yam, plantain, banana, cassava, cocoa, rice, wheat, cowpea, groundnut, millet, maize, sorghum (Paeth, Capo-Chichi & Endlicher, 2008; Jarvis et al., 2012; Nelson et al., 2014; Sultan & Gaetani, 2016). For example, yam production in the region constitutes about 91% of the global production. In Sub-Saharan Africa, SSA, Cassava remains the most important staple food crop in the region in terms of production due to its high resilience to drought (Jarvis et al., 2012; Srivastava, Gaiser & Ewert, 2016; Sultan & Gaetani, 2016). Sorghum and millet also account for about 64% of the cereal production in West Africa (FAOSTAT, 2014; Sultan & Gaetani, 2016). Maize adjudged to be the most important staple food in SSA provides about 20% of the calorie intake in the West African region (FAOSTAT, 2014; Sultan & Gaetani, 2016). Cash crops such as cocoa, oil palm and other crops such as plantain also contribute significantly to the region’s economy.

Data

Historical and future climate datasets

For this study, three data sets were used, namely observations of present-day climate and the locations where crops are grown as observed from crop suitability model; Ecocrop output; modelled simulations of present and projected crop suitability driven by observed and projected climate data. The observation dataset was the 0.5° × 0.5° resolution monthly precipitation and temperature gridded dataset for the period of 1901 to 2016 obtained from the Climate Research Unit (CRU TS4.01, land only) University of East Anglia (Harris et al., 2014). This was used to evaluate the available bias corrected Regional Climate Models, RCMs forced by 10 Global Climate Models (GCMs) from the Coupled Model Intercomparison Project Phase 5 (CMIP5) (Taylor, Stouffer & Meehl, 2012). The regional climate simulation was obtained from the Swedish Meteorological and Hydrological Institute, Linköping, Sweden—Rossby Centre’s regional atmospheric model (SMHI-RCA4, hereafter RCA4) (Samuelsson et al., 2011). The modelled climate data were used as inputs into the crop suitability model, Ecocrop (Hijmans et al., 2001). For this study, six crops were studied—millet and maize (Cereals); cassava and plantain (Root and Tuber); cowpea (Legume); and groundnut (Cash crop). These were selected based on their economic importance in the region. The different data sets are defined in the sub-sections below.

Rainfall and Temperatures are important climate variables used in determining the impacts of climate change at different scales (Cong & Brady, 2012; Mastrandrea et al., 2015) and have a significant effect on crop yield (Abbate et al., 2004; Medori et al., 2012). While rainfall affects crop production in relation to photosynthesis and leaf area, temperature affect the length of the growing season (Olesen & Bindi, 2002; Cantelaube & Terres, 2005). For this study, we used the bias-corrected mean monthly minimum temperature (tmin), mean monthly temperature (tmean) and total monthly precipitation (prec). Data from 10 CMIP5 GCMs downscaled by RCA4 are used as input into crop suitability model (see Table 1). We used Representative Concentration Pathways (RCP) with highest CO2 concentration, RCP8.5 for the analysis to present the influence GWL1.5, GWL2.0, & GWL3.0—on crop growth suitability over West Africa. We used RCP8.5 because it matches the current emission path in CO2 increase and covers the range of three temperatures over the largest number of simulations ensemble members (Abiodun et al., 2019).

Table 1 List of dynamically downscaled GCMs used in the study.

Modelling institution	Institute ID	Model name	Resolution	
Canadian centre for climate modelling and analysis	CCCMA	CanESM2	2.8° × 2.8°	
Centre National de Recherches Meteorolo-Giques/Centre Europeen de Recherche et Formation Avanceesencalcul scientifiqu	CNRMCERFACS	CNRM-CM5	1.4° × 1.4°	
Commonwealth Scientific and Industrial Research Organisation in collaboration with the Queensland Climate Change Centre of Excellence	CSIRO-QCCCE	CSIRO-Mk3.6.0	1.875° × 1.875°	
NOAA geophysical fluid dynamic laboratory	NOAAGDFL	GFDL_ESM2M	2.5° × 2.0°	
UK Met Office Hadley centre	MOHC	HadGEM2-ES	1.9° × 1.3°	
EC-EARTH consortium	EC-EARTH	ICHEC	1.25° × 1.25°	
Institut Pierre-Simon Laplace	IPSL	IPSL-CM5A-MR	1.25° × 1.25°	
Japan agency for Marine-Earth Science and Technology	MIROC	MIROC5	1.4° × 1.4°	
Max Planck institute for meteorology	MPI	MPI-ESM-LR	1.9° × 1.9°	
Norwegian climate centre	NCC	NorESM1-R	2.5° × 1.9°	

Ecocrop dataset

Ecocrop is an empirical model originally developed by Hijmans et al. (2001) from the result of field experiments run across the world a database of crop thresholds based on the FAO-Ecocrop database. It is designed on a monthly scale to project the suitability of a crop in relation to the climate conditions over a geographical area, based on the crop growth threshold dataset from the FAO-Ecocrop database (Hijmans et al., 2001; Ramirez-Villegas, Jarvis & Läderach, 2013). The crop growth threshold varies with crop species, climatic and geographical conditions. These crop thresholds describe the monthly suitability range of plant species against total monthly rainfall (prec.), monthly minimum temperature (tmin), mean temperature (tmean) and maximum temperature (tmax) over the length of its growing season (Dixon, Gulliver & Gibbon, 2001). The computation of optimal to non-optimal conditions based on this data, allows for the simulation of monthly crop suitability in response to monthly climatic variables. Ecocrop computes the relative suitability of a crop in response to climate variables such as total monthly rainfall (prec.), monthly minimum temperature (tmin), mean temperature (tmean) and maximum temperature (tmax) over the length of its growing season thus generating a suitability index score from 0 (unsuitable/non-optimal) to 1 (highly suitable/optimal), see Table 2 (Hijmans et al., 2001; Ramirez-Villegas, Jarvis & Läderach, 2013). It is important to state here that the strength of Ecocrop is in its ability to help understand the spatio-temporal distribution of crop suitability over a large area for a long time period and for multiple crops which makes it suitable for this study. Hence, we don’t expect the model to fit a specific crop over a specific point in space or at field scale as compared to other crop models.

Table 2 A description of Ecocrop suitability index value used for the study (Adapted from Egbebiyi, Crespo & Lennard, 2019).

Suitability index value	Category/Description	
0.0–0.2	Unsuitable	
0.21–0.40	Very Marginally suitable	
0.41–0.60	Marginally suitable	
0.61–0.80	Suitable	
0.81–1.00	Highly suitable	

Ecocrop performs two different calculation using the two climate variables (temperature and rainfall) to calculate the suitability of a crop based on the potential 12 months growing seasons of the year. This assumes the first month of the growing season for each month (i) as described in Ramirez-Villegas, Jarvis & Läderach (2013). Temperature suitability is calculated by comparing the crop parameters with the minimum and mean temperature while the rainfall suitability calculation is done using the crop’s growing season total rainfall. To compute the total suitability index, the suitability score of temperature (Tsuit) and precipitation (Rsuit) which have been calculated separately (see Hijmans et al. (2011); Hijmans et al. (2017)) for more information on the suitability computation) are then multiplied together as shown below: SUIT=Tsuit∗Rsuit.

Where—SUIT is the total crop suitability index.

Rsuit is the rainfall suitability score.

Tsuit is the Temperature suitability score.

A 12-month suitability index output is generated, with each month showing the most suitable conditions for the crop. The description of the suitability index values follows that used in Egbebiyi et al. (2019).

Ecocrop model can be used as a tool for climate and crop suitability assessments, strategic spatial, seasonal and temporal planning for crop production. According to Jarvis et al. (2012), the suitability rating of the model offers some relation to agricultural yield, however this relationship is often difficult to capture (Ramirez-Villegas, Jarvis & Läderach, 2013). As a result, the model has been employed in the suitability projection of several crops such as cassava (Jarvis et al., 2012; Egbebiyi et al., 2019; Egbebiyi, Crespo & Lennard, 2019), sorghum (Ramirez-Villegas, Jarvis & Läderach, 2013; Kim et al., 2018; Egbebiyi, Crespo & Lennard, 2019), yam (Egbebiyi et al., 2019; Remesh et al., 2019), maize and banana (Hunter & Crespo, 2018), sugar cane (Abdallah, 2018; Abdallah & Jaafar, 2019) and other crops (Beebe et al., 2011; Ramirez-Cabral, Kumar & Shabani, 2017; Egbebiyi et al., 2019).

However, we clearly acknowledge the numerous other environmental factors that contribute to crop growth suitability namely climate, water, soil, farm management ad typology, nutrients, pests and disease etc. However, the present study only focuses on the impact of climate on crop suitability growth in response to climate variables, monthly minimum and mean temperature and total monthly rainfall based on Ecocrop developed by the Food and Agriculture Organization (FAO). Nevertheless, other factors such as soil type, farm management can also influence crop suitability over the region but are not considered in the present study due to the model parameter which is a limitation of the Ecocrop model.

Although we understand that those thresholds will vary depending on crop varieties or location, the concept and the general validation of the thresholds makes it a suitable tool to assess different crop’s suitability over large areas. Previous studies have reported a good agreement between climate change impacts projections from Ecocrop model and other crop models (Ramirez-Villegas, Jarvis & Läderach, 2013; Vermeulen et al., 2013; Challinor et al., 2014; Rippke et al., 2016). It should be noted that this study did not undertake any additional ground-truthing or calibration of the range of climate parameters preferred for the crops, and therefore the default Ecocrop parameters were assumed to be suitable. We therefore used this approach to evaluate crop suitability during the historical period and under different GWLs.

Simulation approach

We calculated the time of reaching the 1.5, 2 and 3 °C temperature warming over West Africa using RCP8.5 emission and a baseline period of 1971–2000. We calculated the year of reaching 1.5, 2 and 3 °C of global warming under RCP8.5 using the method of Déqué et al. (2017) and Nikulin et al. (2018). A thirty-year running average was used to calculate the mid-year in which each global warming level (GWL) is reached relative to the pre-industrial baseline period of 1861–1890. The timing of reaching 1.5, 2 and 3-degree is projected as 2025, 2038, 2048 respectively (Nikulin et al., 2018). All the extracted and downscaled CMIP5 datasets by RCA4 were bias corrected with the observation based reference data WATCH-Forcing-Data-ERA-Interim (WFDEI) dataset (Weedon et al., 2014). This is crucial because regional climate models often deviate from the observed climatological data hence the need for bias correction before the data is used for climate change impacts assessment such as hydrological modelling and agricultural impact studies (Chen, Brissette & Lucas-Picher, 2015; Vrac, Noël & Vautard, 2016; Famien et al., 2018). We evaluated the bias corrected RCA4 historical data against the CRU dataset. The results showed that there is a good agreement between observation dataset (CRU) and the bias corrected RCA4 monthly simulated past climate data for both temperature and precipitation over West Africa. RCA4 bias-corrected output has a strong correlation (r ≥ 0.8 and r ≥ 0.6) with the CRU datasets for temperature and total monthly rainfall datasets respectively. For example, the model replicates the CRU north-south temperature gradient, concurring with past studies (Gbobaniyi et al., 2014). RCA4 simulated total monthly rainfall realistically captures the essential features namely, both the zonal pattern and meridional gradient and the rainfall maxima over high topography (i.e., Cameroon Mountains and Guinean Highlands) as observed in CRU. Agreeing with previous findings (Egbebiyi, 2016; Klutse et al., 2016; Abiodun et al., 2017). The performance of RCA4 in simulating the essential features of West African climate variables, temperature and rainfall, makes it suitable and gives confidence in the use of the RCA4 for crop suitability simulation over the region. Also, the use of Ecocrop was based on past finding by Egbebiyi et al. (2019) that there is a good agreement between Ecocrop and MIRCA2000 data set, a global monthly gridded data of annual harvested area around year 2000 (Portmann, Siebert & Döll, 2010) for the different crops. The study showed a strong spatial correlation (r > 0.7) for the examined crops in this study between Ecocrop and MIRCA2000 simulation. This gives some level of confidence in the use and performance of the Ecocrop simulation over the region.

The influence of the GWL1.5, 2.0 and 3.0 crop suitability and month of planting was assessed based on the methodologies described in Ramirez-Villegas, Jarvis & Läderach (2013). The resulting tmin, tmean and prec values from the 10 downscaled GCMs over the 30-year window at the time of reaching the 1.5, 2 and 3 °C GWLs were calculated and used as input data into Ecocrop model to compute the suitability index for each crop across over West Africa. The results were then used to assess how each GWL will impact crop suitability across the Agro-Ecological Zones (AEZs) of West Africa. After the simulation, we computed the mean of the best three consecutive suitability index and best three months of planting window within the growing season across each grid point over the region for the historical and future analysis for the three GWL warming levels. This was done to remove the influence of the unsuitable and marginally suitable months from the averaged suitability spatial distribution within a growing season and varies for each crop. The contour lines represent the regions with marginal to highly suitable mean crop suitability over West Africa over the historical period.

Assessing the robustness of climate change

We assessed the robustness of the projected climate change via the three GWLs based on two conditions. Firstly, at least 80% of the simulation must agree on the sign of change. Secondly, at least 80% of the simulations must indicate that influence of climate change is statistically significant, at 99% confidence level using a t test with regards to the baseline period, 1971–2000. When these two conditions are met then we consider the climate change signal to be significant. Previous studies (Abiodun et al., 2019; Klutse et al., 2018; Maúre et al., 2018; Nikulin et al., 2018) have all used the methods to test and indicate the robustness of climate change signals. We also assess the trend of change in crop suitability and month of planting at each global warming levels for each crop using Theil-Sen estimator or Sen’s slope (Theil, 1950; Sen, 1968). The Theil-Sen slope estimator is non-parametric and applied in the estimation of magnitude of trend. It is more robust such that it is less sensitive to outliers in the time series as compared to standard linear regression trend (Wilcox, 2001). Theil-Sen slope method can detect significant trends with changing rate than the linear trend (Ohlson & Kim, 2015). Previous studies (Wilcox, 1998; Peng, Wang & Wang, 2008) have used the method in calculating trends.

Results

Simulated crop suitability in the historical climate over West Africa

RCA4 simulated crop suitability from observed climatology inputs (CRU-Ecocrop) shows a decreasing mean suitability from south to north of West Africa (north-south suitability gradient) (Figs. 2 and 3, column 1). The spatial suitability representation reveals unsuitable or very marginal suitability to the north in the Sahel from latitude 14°N with a low Suitability Index Value (SIV) value between 0.0 and 0.4. and higher suitability to the south in the Guinea-Savanna AEZ with a high SIV (0.6–1.0) sandwiched by an ash/silver suitability line called the Marginal Suitability Line (MSL) with SIV between 0.41 and 0.59. In general, MSL are observed around latitude 14°N in the Sahel AEZ (northern Sahel) for the simulation across the region except for the one observed around latitude 12°N boundary between the Sahel and Savanna AEZ. Ecocrop simulation of the crop types examined, legumes (cowpea and groundnut), root and tuber (cassava and plantain) and cereals (maize and pearl millet) are very suitable to the south of the MSL with no or low suitability to the north. Along the coastal areas, legumes and root and tuber crops are suitable along the south-west coast of Senegal to the south-west coast of Cameroon. For cereals, pearl millet is suitable along the west coast of Senegal and from the south coast Ivory Coast to the south coast of south-west coast of Cameroon while maize is suitable from the south coast of Ivory Coast to the south-west coast of Nigeria.

Figure 2 The spatial distribution of crop suitability as simulated by Ecocrop over West Africa for Hist. (column 1) and (column 2–4) at different global warming levels (GWL1.5, GWL2.0, GWL3.0) under RCP8.5 for cassava, cowpea and groundnut.

The white areas along the coast do not have data. (0.0 > not suitable > 0.2 > very marginal > 0.4 > marginal > 0.6 > suitable > 0.8 > highly suitable). The contour lines represent crop suitability in the historical climate. The vertical strip (|) indicates where at least 80% of the simulations agree on the sign of the changes, while horizontal strip (−) indicates where at least 80% of the simulations agree that the projected change is statistically significant (at 99% confidence level). The cross (+) shows where both conditions are satisfied; hence, the change is robust.

Figure 3 Spatial distribution of crop suitability over West Africa as simulated by Ecocrop for Hist. (column 1) and column (2–4) at different global warming levels (GWL1.5, GWL2.0, GWL3.0) under RCP8.5 scenario for maize, pearl millet and plantain.

The white areas along the coast have no data. (0.0 > not suitable > 0.2 > very marginal > 0.4 > marginal > 0.6 > suitable > 0.8 > highly suitable). The contour lines represent crop suitability in the historical climate. The vertical strip (|) indicates where at least 80% of the simulations agree on the sign of the changes, while horizontal strip (−) indicates where at least 80% of the simulations agree that the projected change is statistically significant (at 99% confidence level). The cross (+) shows where both conditions are satisfied; hence, the change is robust.

Ecocrop was also used in simulating the best planting months (PM) from range of month in a planting window within the Length of Growing Season (LGS) over West Africa for the historical climate (Figs. 4 and 5, column 1). LGS provides information on the start and end of growing season and can also assist in the simulation process of identifying the best PM within a possible planting window in a growing season over given location. The simulated planting month represent the first month of the best three months of the planting window and varies with crop types across the three AEZs of the region i.e., a simulation of April means April–June is the three best PM. For the legumes, our simulation shows January–July as the planting windows for legume crops, cowpea and groundnut over the region, but Jan (Jan–Mar) and Feb (Feb–April) as the three-best PM for cowpea and groundnut respectively for large part of the region in the central Guinea and Savanna AEZs except over Sierra Leone, Liberia and south coast of Nigeria. The month of Feb. (Feb–April) was simulated as the best three-month planting period in western and eastern Savanna-Sahel AEZs for cowpea while it was Mar. (March–May) over the same area and period for groundnut. Along the coastal areas, July is simulated as the PM along the southwest coast of southern Sierra Leone to Liberia and the south coast of Nigeria and April along the southwest coast of northern Sierra Leone. For Groundnut April, is PM along the west coast of Guinea, May along the west coast of Sierra Leone and northern Liberia. August and March at south coast of Liberia and Nigeria respectively. The months of December and January are the PMs along the south coast of Ivory Coast to Ghana for cowpea and groundnut respectively.

Figure 4 Spatial distribution of the best three planting months as simulated by Ecocrop over West Africa for Hist. (column 1) and (column 2–4) at different global warming levels (GWL1.5, GWL2.0, GWL3.0) under RCP8.5 for cassava, cowpea and groundnut.

The colour in the historical month represents the first month of the best three consecutive months (e.g., a simulated planting month showing September means September–November planting period). The green and brown colour shows projected delay and early shift in the planting month from the historical climate. The vertical strip (|) indicates where at least 80% of the simulations agree on the sign of the changes, while the horizontal strip (−) indicates where at least 80% of the simulations agree that the projected change is statistically significant (at 99% confidence level). The cross (+) shows where both conditions are satisfied; hence, the change is robust.

Figure 5 Spatial distribution of the best three planting months as simulated by Ecocrop over West Africa for Hist. (column 1) and (column 2–4) at different global warming levels (GWL1.5, GWL2.0, GWL3.0) under RCP8.5 for maize, pearl millet and plantain.

The colour in the historical month represents the first month of the best three consecutive months (e.g., a simulated planting month showing September means September–November planting period). The green and brown colour shows projected delay and early shift in the planting month from the historical climate. The vertical strip (|) indicates where at least 80% of the simulations agree on the sign of the changes, while the horizontal strip (−) indicates where at least 80% of the simulations agree that the projected change is statistically significant (at 99% confidence level). The cross (+) shows where both conditions are satisfied; hence, the change is robust.

Root and tuber crops; plantain is an annual crop that can be planted in any month of the year (Figs. 4 and 5, column 1). The simulated PM is an overlay of the simulation of other months in the year as the crop may be planted in the suitable zones, Guinea and Savanna at any month/period of the year. For cassava, our simulation shows March (March–May) as the best PM generally over the region (Guinea-Savanna AEZs) except along the south-east coast of Ivory Coast to Ghana with PM in August, northern Guinea to Gambia and south east Senegal as well as the boundary of Benin Republic to north west Nigeria with PM in April. Our simulation for cereals shows February as PM for millet in the Guinea and March, April in the Savanna and Sahel AEZs respectively although there are exceptions. For example, in the central Savanna, from northern Benin Republic to north-western Nigeria, pearl millet PM is April while in the north-eastern Nigeria in the Sahel it is March compared to April in the Sahel zone. However, pearl millet PM is April in the western Sahel along the south-west coast of Senegal, June along the west coast of Guinea and January along the south coast of Ivory Coast to the south-west coast of Nigeria. Maize PM is simulated to be in May (May–July) in the Guinea and southern Savanna zone of West Africa while it is in December (December–February) in the northern Savanna into the Sahel zone.

These evaluation simulations (RCA4-Ecocrop) captures the observed variation in suitability distribution over a large-scale area for the different crops across the three AEZs of West Africa in the present-day climate when compared to MIRCA2000 gridded global datasets. This serves as a baseline for evaluating the changes in crop suitability under global warming levels of 1.5 to 3 °C over the region. The model also captures the growing season of crops over the region which varies with different months of the year.

Projected changes in crop suitability under different GWLs 1.5, 2.0. 3.0 over West Africa

At all warming levels, Ecocrop projects a similar spatial suitability distribution pattern in crop suitability over West Africa (Fig.2, column 1). For instance, projected spatial suitability distribution under the three warming levels show a similar pattern of decreasing suitability index value (SIV) from south to north over West Africa with high and low suitability to south and north respectively. For all the GWLs, there is no projected latitudinal shift from 14°N (north of the Sahel AEZ) and 12°N (north of the Savanna AEZ) in the marginal suitability area as observed in the historical climate. The projected spatial suitability distribution under all the GWLs show higher SIV (0.6–1.0) remains in the Guinea-Savanna zone which is to the south of the marginal suitability while low SIV (0.0–0.4) are to the north of the MSL as observed in the historical climate. Similarly, projected suitability pattern remains similar along the coastal areas under the three global warming levels as for the historical climate.

Ecocrop projected change in crop suitability vary for different crop types at all warming levels. However, the magnitude of the projected change varies over the region and increase with increasing GWLs (Figs. 2 and 3, column 2 and Table 3). The change in SIV means an increase or decrease in the suitability index value of crop of one AEZ and GWL. For example, a 0.1 SIV increase for a crop with SIV 0.4 (in the past climates) under GWL2.0 means an increase in SIV 0.5 and a change from very marginal suitable area to being marginally suitable under GWL2.0. At GWL1.5 (Figs. 2 and 3, column 2 and Table 3), For legume crops, cowpea and groundnut, projected suitability change is over the central Savanna AEZ (from the northeast Ivory Coast to northeast Nigeria) extending to the southern Sahel with a magnitude increase of 0.1 except over the south-western area of Chad Republic, which is east of southern Sahel. The projected change shows the suitability of legumes from very marginal to being marginally suitable in the southern Sahel. Generally, no change in suitability is projected over the Guinea AEZ and over the western and eastern Savanna except in the coastal areas. No change in crop suitability is also projected north of 14°N under GWL1.5. However, some areas with pockets of projected suitability decrease (SIV = −0.1 under GWL1.5) are observed in the southern part of Nigeria and south-western part of Sierra Leone for cowpea. Along the coastal area projected decrease in crop suitability is projected along the south-west coast of Sierra Leone and the south coast of Nigeria for cowpea and groundnut respectively. For cereals, maize and pearl millet, a projected increase about 0.2 in SIV is expected in the central Sahel under GWL1.5 for pearl millet and 0.1 SIV increase over the Sahel for maize around 12–14°N and central Savanna. However, despite the projected crop suitability increases in the Sahel, the cereal crops will only be marginally suitable for cultivation except over the Savanna AEZ. Also, pockets of crop suitability increase are projected in north eastern part of Nigeria, south of Burkina Faso in central Savanna and along the south coast of Ivory Coast in the Guinea for maize. In contrast, south of 14°N no change in suitability under GWL1.5 is projected, but with some exceptions along the coastal areas of Guinea and Nigeria in Savanna and Guinea AEZs. Over the coastal areas, decreases in crop suitability about -0.2 are projected in the south coast of Nigeria and along the west coast of Guinea and Sierra Leone. A decrease of similar magnitude is also projected in the north east boundary of Nigeria and Cameroon and in the central and north-western parts Nigeria in the Guinea and Savanna AEZ respectively for both crops. Under GWL1.5, for root and tubers, suitability increases about 0.1 is projected over the central Savanna while a similar magnitude decrease is projected west and eastern Savanna for cassava. Plantain is projected to decrease in suitability (about −0.1) in the Guinea zone except along the southeast boundary between Nigeria and Cameroon with a projected suitability increases about 0.2 as in the central Savanna. The projected change in crop suitability under GWL1.5 is robust, in that at least 80% of the simulation agree with sign of change and that the projected change in suitability are statistically significant (at 99% confidence level) for all the crop types.

Table 3 Projected changes in crop suitability over West African AEZs at different global warming levels.

Crops	GWL1.5	GWL2.0	GWL3.0	
	Guinea	Savanna	Sahel	Guinea	Savanna	Sahel	Guinea	Savanna	Sahel	
Cassava	No change remains suitable	No change, remains suitable	No change, very marginally suitable	A 0.1 SIV decrease, remains suitable	A 0.1 SIV decrease, still suitable	A 0.1 SIV decrease becomes unsuitable	A 0.2 SIV decrease but still suitable	A 0.2 SIV decrease but still suitable	Above 0.2 SIV decrease becomes unsuitable	
Cowpea	No change, highly suitable	A 0.1 SIV increase, highly suitable	A 0.1 SIV increase in the southern Sahel, marginally suitable	No change, highly suitable	Same as GWL1.5	Same as GWL1.5	No change in suitability	Same as in GWL1.5	Same as in GWL1.5	
Groundnut	No change in suitability	No change in suitability	Same as Cowpea	Same as in GWL1.5	Same as GWL1.5	Same as GWL1.5	Same as in GWL1.5	Same as GWL1.5	Same as GWL1.5	
Maize	Very suitable except the coastal areas of Nigeria and Liberia	Very suitable except Sierra Leone and west coast of Guinea	A 0.1 SIV increase now suitable in the south Sahel	Same as in GWL1.5	About 0.1 SIV decrease but still suitable	Same as GWL1.5	No change in suitability	About 0.2 decrease in SIV but still suitable	Same as GWL1.5	
Pearl millet	Very suitable except the south coast of Nigeria	No change in SIV	No change but about 0.1 SIV increase in central Sahel	No change with about 0.1 SIV decrease east Guinea	About 0.1 decrease in SIV but still suitable	Same as GWL1.5	Same as GWL2.0	About 0.2 decrease in SIV but still suitable	A 0.2 SIV decrease & increase in western and central Sahel respectively	
Plantain	A 0.1 SIV decrease but still suitable	A 0.2 SIV increase, now suitable in the savanna zone	No change in suitability remains unsuitable	Same as GWL1.5	Same as GWL1.5	Remains Unsuitable	A 0.1 SIV decrease but still suitable	Same as GWL1.5	Remains unsuitable	

Under GWL2.0, the impact of the warming on crop suitability shows a similar spatial suitability pattern as GWL1.5 over West Africa, but with an intensification of GWL1.5 effect across the different crop types over the region (Figs. 2 and 3, column 3 and Table 3). The intensity of change at GWL2.0 warming in comparison to GWL1.5 are most drastic on cereals and root and tuber crops compared to the legumes both in magnitude of change and projected spatial suitability distribution. The meridional (N-S) movement via projected increase (expansion) and decrease (contraction) in magnitude and spatial suitability distribution at different GWLs shows contraction is mainly to the south (around 14°N, marginal suitability line from the historical climate and 0.4 contour line, 0.4 marginal suitability line) and expansion to the north of the root and tuber and cereal crops except maize. As seen from Fig. 2, cassava remains the most impacted crop in the region as a delta 0.5 °C temperature further reduces areas suitable for cultivation of the crop over West Africa. A reduction in suitable areas are also projected for groundnut and maize south of 14°N although majorly with maize and in the eastern Sahel for cowpea in the south western area of Chad. On the other hand, the 2 °C warming may also lead to an expansion in suitability over the region. A projected spatial increase through an expansion of suitable areas for crop types except cassava is expected at GWL2.0. The projected suitability increase has a similar spatial pattern as GWL1.5 but with an increased magnitude of change in the suitability index value. All the projected change at GWL2.0 is robust (i.e., statistically significant at 99% confidence level and 80% of the model agree with the sign of suitability change). The increase in the reduction of suitable areas over the regions, notably with cereals and root and tuber crops at GWL2.0 suggests keeping global warming to 1.5 °C may limit decrease in projected SIV and spatial suitability of the affected area within the natural variability of the reference/historical climate.

The impact of the increase in global warming beyond GWL1.5 and GWL2.0, will be more drastic on cereals and root and tubers with GWL3.0 over West Africa (Fig. 5, column 4, see also Table 3). Under GWL3.0, the spatial suitability distribution over the region shows a similar spatial suitability distribution pattern over the region as the historical climate with higher suitability to the south of MSL around latitude 14°N and low suitability to the north of the region. Projected change in SIV and suitable areas show an increase in the intensity of change with increased warming compared to GWL1.5 and GWL2.0 especially the cereal and root tuber crops. For example, projected decrease of 0.1 and up to 0.3 of SIV is projected for cassava along the coastal areas and further inland respectively across the three AEZs. The projected SIV decrease for cassava under GWL3.0 will result in a decrease in suitable areas from the northern Savanna to south of the Sahel around 14°N zone except the coastal area in Savanna. The projected decrease in crop suitability over these areas shows the northern Sahel and southern Savanna will become unsuitable and marginally suitable respectively, for the cultivation of cassava under GWL3.0 except in the southern Savanna and Guinea zone compared to the historical period thus showing a constraint in growing the crop only in the southern area of the region. Plantain will also experience decreases up to 0.3 SIV over the Guinea zone and along the western area of the Savanna however, the crop remains suitable over the area. On the other hand, an increase in SIV above 0.2 is expected from the western to eastern Savanna except over the north-central area of Nigeria. The projected SIV increase also means an increase in the suitable area for the cultivation of plantain with increased suitability from marginally suitable to being suitable. This projected expansion in suitable areas for plantain thus provide an opportunity of more area for cultivation of the crop. As seen in Fig. 3 and described in Table 1, both maize and pearl millet under GWL3.0 remains suitable over the Guinea and Savanna zone despite the projected decrease in SIV. However, the good news is the spatial increase in suitable area for the crop in central Sahel as compared to being marginally suitable in the past climate, thus expanding northward in suitable areas which may be improve the production of the crop. There are not much changes in the SIV and suitable areas for the cultivation of legume crop, cowpea and groundnut under GWL3.0.

Conversely under GWL3.0, projected decrease in SIV due to an increased warming will lead to a further decrease in crop suitability of most suitable areas in the past climate. This is particularly expected over the Guinea and Savanna zones for roots and tuber and cereal crop types due to a decrease in magnitude of SIV and spatial contraction in suitable areas of these crop types. Also, it is what stating that the projected changes under GWL2.0 and GWL3.0 are robust (i.e., are statistically significant at 99% confidence level) for all the crop types. In addition, as mentioned with GWL impact warming above, a projected spatial crop suitability change may not change the crop’s suitability spatial distribution status (e.g., from unsuitable to suitable or may remain marginal or highly suitable) due to an increase warming over West Africa. For example, despite the projected decrease in suitability index magnitude for cassava across West Africa, the crop will remain very suitable south of 14°N under GWL2.0 and GWL3.0 warming over the region. On the other hand, despite the projected increase in suitability for groundnut, cowpea and maize north of 14N over west Africa, these crops still retain the unsuitable to marginally suitable characteristics in the Sahel zone.

Impact of different GWLs on crop planting period/month over West Africa suitability

At all global warming levels, Ecocrop projected change in the planting period/month varies for different crop types across the different AEZs of West Africa (Figs. 4 and 5, column 2–4 and Table 4). The increased warming resulted in early or late/delay in PM for different crops and increases in magnitude with increasing warming level. It is worth stating that the change in PM describe a change in the best three planting months under the three GWLs. For example, under GWL1.5 no change in PM is projected for legume crops except over the Sahel (around 13°N) and along the coastal area (Fig. 4, column 2–4, Table 2). A one-month delay in the PM (Feb to March) is projected in the Sahel for both Cowpea and Groundnut as compared to the past climate. Along the coastal area, about two-month delay in the PM is projected along the south-west coast from Sierra-Leone to Liberia and up to 3–4 months extending to the south coast of Ivory Coast for Cowpea. A similar magnitude of delay in PM as Cowpea is projected for Groundnut along the south-west coast from Sierra Leone to Liberia except in the north-east of Sierra Leone. Under GWL1.5, early planting of about one-month PM (i.e., from February to January) is projected in the south-east of Nigeria for Cowpea as compared to the past climate while Groundnut, a similar one-month early planting (February-January) is projected in the north-east of Sierra Leone. Also, a two-month early planting along the south coast of Nigeria is predicted under GWL1.5 (i.e., February to December of the preceding year). This means there is a shift in the PM from February-April in the past climate into December-February under GWL1.5.

Table 4 Projected changes in time of planting (crop planting months) over West African AEZs at different global warming levels.

Crops	GWL1.5	GWL2.0	GWL3.0	
	Guinea	Savanna	Sahel	Guinea	Savanna	Sahel	Guinea	Savanna	Sahel	
Cassava	Delayed planting for two months	Early planting by four months	One delay in southern Sahel zone	Same as GWL1.5	Same as GWL1.5 but for more area	No planting, date	Same as GWL1.5	Same as GWL1.5	No planting date	
Cowpea	One month delayed planting	No change in planting date	No change in planting date	Same as GWL1.5	Same as GWL1.5	No change in planting date	Same as GWL1.5	Same as GWL1.5	No change in planting date	
Groundnut	On month delayed planting	No change in planting date	No change in planting date	Same as GWL1.5	Same as GWL1.5	No change in planting date	Same as GWL1.5	Same as GWL1.5	No change in planting date	
Maize	Three months delayed planting	Four months early & delay planting in east & west respectively	No change in planting date	Same as GWL1.5	Same as GWL1.5	No change in planting date	Same as GWL1.5	Same as GWL1.5	No change in planting date	
Pearl millet	One month delayed planting	Two months delayed planting	Two months delayed planting	Same as GWL1.5	Same as GWL1.5	Same as GWL1.5	Same as GWL1.5	Same as GWL1.5	Same as GWL1.5	
Plantain	No change in planting date	No change in planting date	No change in planting date	No change in planting date	No change in planting date	No change in planting date	No change in planting date	No change in planting date	No change in planting date	

For cereal crops (Fig. 5, column 2–4, Table 4), a general delay in the PM is projected across under GWL1.5 except over Sierra Leone and its boundary south-east boundary with Liberia and north-east boundary with Guinea as well as the central Guinea-Savanna zone in Nigeria except the south coast for millet and in the western and eastern Savanna for maize. Projected delays in the PM for millet is about two months across the region and may be about four months in the central Sahel zone, south of Sierra Leone and south coast of Nigeria under GWL1.5. Delay about two months in the PM is projected for maize from Ivory Coast to central Cameroon in the Guinea zone, while the delay in PM is projected to be above four months from the central part of Nigeria extending to its boundary in the north with the Niger Republic in the central Savanna-Sahel AEZ and in the south of Chad Republic in the south-eastern Sahel. Conversely, under GWL1.5, an early planting about 3–4 months (i.e., from December during the past climate to August) is projected from the east of Guinea extending to western Nigeria in the Savanna-Sahel AEZs and along the north boundary of Cameroon and South of Chad Republic in the eastern part of the Savanna AEZs for maize. For Millet, early planting about 1–2 months (from February to December) is projected in the Savanna zone from Sierra Leone and its boundary south-east boundary with Liberia and north-east boundary with Guinea and from the south to north-central part of Nigeria in central Guinea-Savanna zone except the south coast in the coastal area.

Projected changes in root and tuber crops follow a similar pattern under the three GWLs although with different magnitudes at different warming level and crop (Figs. 4 and 5, column 2–4, Table 4). For example, about 2 months delay in planting of cassava in the Guinea zone and the western Savanna zone under the three global warming levels. The projected change means a change in the planting date of cassava from March to May along the west coast of Guinea (western Savanna) to the south-west coast of Liberia and from the south coast of Nigeria to the southern Cameroon (Guinea zone). Along coastal areas, the projected change in PM is from June to August from the south coast of Ivory Coast to Ghana in the Guinea Zone. Additionally, under GWL1.5 a delay in PM of similar magnitude is predicted in the north-east Nigeria, along the south-west coast from Senegal to Guinea and from the south-east Mali to the central region of Burkina Faso in the Sahel. On the other hand, an early planting is projected for cassava in the central Savanna zone from the south east Mali to the south of Chad Republic in the eastern Savanna zone except in the north-east Nigeria in the eastern Savanna under GWL1.5. The projected change in PM is about 4 months earlier (April to December), compared to the past climate in the Savanna zone. For plantain, no change in PM is projected under GWL1.5. The no change in the month of planting may be linked to it be an annual crop which can be planted at any month in the year.

At GWL2.0, projected change in crop PM show a similar spatial characteristic in projected crop PM change across the three AEZs and crop types as simulated under GWL1.5 except with few discrepancies in some areas or crop types (Figs. 4 and 5, column 3, Table 4). Legume crops projected change in PM under GWL2.0 show similar spatial pattern for both delay and early in PM across the region as GWL1.5 except for Groundnut in the south coast of Nigeria. An additional 0.5 °C of warming is projected to potentially lead to an early planting of Groundnut in south coast of Nigeria about 2-3 months, PM December, under GWL2.0 compared to the PM in February in past climate i.e., a change in PM from February to December of the preceding year. For cereals, projected change in PM similar spatial pattern for both maize and pearl millet under GWL2.0 as projected under GWL1.5 except for an increase in magnitude in the projected PM in the central Sahel under GWL2.0 for Pearl millet. A 2-month late/delayed planting is projected in southern Niger Republic in the central Sahel zone under GWL2.0. The projected delay means a change in the PM from April (April–June) in the past climate to June (June–August) under GWL2.0 as compared to the one-month delay under GWL1.5 over the area. This suggests limiting the global warming to 1.5 °C may help maintain the in planting and cultivating period over this area within the natural variability of the reference climate. Projected PM under GWL2.0 shows a delay about 1-2months to the south in the Guinea zone and along the west coast from Guinea to south coast of Nigeria for cassava. An early planting of the crop is projected in the north from the southern Senegal in western Sahel Zone to the south of Chad Republic in the eastern Savanna zone. The projected change in PM is about 4 months early (April to December) compared to the past climate in the Savanna zone. Under GWL2.0, no projected change is predicted for Plantain as stated under GWL1.5. All the above projected changes in PM under GWL2.0 are robust for all the crop types.

The projected change in planting month under GWL3.0 show a similar spatial characteristic as that of GWL1.5 and GWL2.0 except for pearl millet over the Sahel zone (Figs. 4 and 5 column 4, Table 4). The increase in warming do not really influence the month of planting under GWL3.0 differently from other warming levels. The main difference in PM under GWL3.0 compared to other GWL1.5 & 2.0 is observed in central Sahel zone for cereal crop, pearl millet. It is projected that Pearl millet is projected will experience a delayed planting about four months compared to the historical climate.

The effect of GWL2 & 3 warming in comparison to GWL1.5 are more drastic on millet and plantain. This is so because the major meridional (N-S) movement via expansion and contraction in suitability due to the increased warming are more observed with the two crops but more with millet. Cassava remains to be the most impacted crops in the region as a 2 °C temperature warming leads to more contracted area in the cultivation of the crop over West Africa. A reduction in suitable areas are also projected for groundnut and maize south of 14°N although majorly with maize and in the eastern Sahel for cowpea in the south western area of Chad. On the other hand, the 2 °C warming may also lead to an expansion in suitability over the region. An intensification of a projected meridional expansion of suitable areas for cowpea, groundnut, maize, millet and plantain in comparison to the 1.5 °C warming are expected coupled with a zonal expansion (E-W spatial movement) in suitability are projected in the Sahel. In addition, as mentioned in the 1.5-degree impact warming above, a projected spatial suitability change may not change the crop suitability spatial distribution status (e.g., from unsuitable to suitable or may remain marginal or highly suitable) due to an increase warming over West Africa. For example, despite the projected decrease in suitability index magnitude for cassava across West Africa, the crop will remain very suitable south of 14°N under GWL2.0 and GWL3.0 warming over the region. On the other hand, despite the projected increase in suitability for groundnut, cowpea and maize north of 14N over West Africa, these crops still retain the unsuitable to marginally suitable characteristics in the Sahel zone.

Trends in projected change in crop suitability and month of planting under different warming levels

We use the Theil-Sen estimator to assess the trends in crops suitability growth for each across the three warming levels over West Africa (Table 5). The trend describes the rate of increase and decrease of the suitable area and SIV with increasing global warming levels. In general, the trends are all positive and the number represents the magnitude of the trend between the projected change in suitability and past climate. Our result shows that there is an increasing trend in crop suitability with increasing warming levels across all the crop types except legumes. Cassava has the highest trend values with an increasing trend value above 0.100 between GWL2.0 and GWL3.0 compared to the 0.028 for GWL2.0 and GWL1.5. The increase in trend value of cassava shows how the crop has been greatly affected by the increasing warming especially under GWL3.0 compared to other crops which has resulted in the loss of suitable areas in cultivating the cassava over the northern Savanna zone to the southern Sahel zone of West Africa. In general, our finding shows the trend value for each crop with each global warming level are almost three-four times the trend value between GWL1.5 and GWL3.0 except cowpea while is farfetched from that projected trend value for GWL1.5 and GWL2.0 is about an average 0.05 except for cassava with almost 0.03 trend increase. This result further confirms the reason we need to strive to ensure we limit global warming to 1.5 °C and to call our scientist and policymakers to the devastating impact of warming of 3 °C on the different crops when compared to GWL2.0. This is in line with Rogelj et al. (2013) that although the NDCs leads to significant reduction in emission, however their impact is below the necessary emission reduction consistent with 2 °C and 1.5 °C climate target. For the Legume crops, there was no change in trend for cowpea and groundnut for all the warmings levels and from GWL1.5 to GWL2.0 respectively while an increasing trend of change in Groundnut is expected with a delta 1.5 °C or a warming beyond GWL2.0, GWL 3.0 over West Africa. On the other hand, there was no trend observed between the month of planting and all warming levels as the trend value shows 1.0 for all the crops and across the three warming levels (see Table 6).

Table 5 Trends in projected changes of crop suitability over West Africa at different warming levels: GWL1.5, GWL2.0, GWL3.0.

Crop	GWL1.5	GWL2.0	GWL3.0	
Cassava	1.026	1.054	1.157	
Cowpea	1.000	1.000	1.000	
Groundnut	1.000	1.000	1.004	
Maize	1.005	1.011	1.036	
Pearl millet	1.006	1.012	1.027	
Plantain	1.000	1.008	1.042	

Table 6 Trends in projected changes of planting month over West Africa at different warming levels: GWL1.5, GWL2.0, GWL3.0.

Crop	GWL1.5	GWL2.0	GWL3.0	
Cassava	1.000	1.000	1.000	
Cowpea	1.000	1.000	1.000	
Groundnut	1.000	1.000	1.000	
Maize	1.000	1.000	1.000	
Pearl millet	1.000	1.000	1.000	
Plantain	1.000	1.000	1.000	

Discussion

Sensitivity of different crop types to different global warming levels in West Africa

The crops examined in this study do not respond homogeneously to global warming. Of the three crops types, root and tubers are most negatively impacted in comparison to cereals and legumes. Root and tuber crops (cassava and plantain) are one of the six most important food crops in the world and cassava is an important staple crop in West Africa (Jarvis et al., 2012; Sultan & Gaetani, 2016). From our findings, spatial contraction and decrease in SIV suitability are projected for both plantain and cassava, in the Guinea and Savanna AEZs under GWL1.5 and GWL2.0. However, under GWL3.0 cassava will no longer be suitable for cultivation in the southern Sahel and to the north of the Savanna zone (between 10–14°N). This may be detrimental for food security and trade as cassava is one of the most important cultivated crops in the region as it can be processed into different product being consumed by the inhabitants of the region (Thiele et al., 2017). In contrast to root and tubers, a spatial expansion into the Sahel AEZ and increased suitability is projected for the legumes (groundnut and cowpea) under the three global warming levels. This may result in increased yield for crops like groundnut which agree with previous findings (Sultan & Gaetani, 2016; Parkes et al., 2018). For the cereal crops (millet and maize), sensitivity to global warming level results in more northward expansion, but with a corresponding spatial contraction for both crops under GWL1.5 and 2.0. An increased intensity in the spatial contraction and loss of suitable areas for the cultivation of the crop is expected under GWL3.0. The spatial expansion northward may be as a result of a projected wetter Sahel (Nicholson, 2013). The projected change in PM of maize corresponds to the main rainy season in the Savanna-Sahel zone. This might be linked to the projected increase in suitability of maize in this zone.

An additional 0.5 °C (GWL2.0–GWL1.5) and 1.5 °C (GWL3.0–GWL1.5) warming- leads to both spatial expansions and contractions of suitability in specific regions and influence the time of planting, early and delay planting over West Africa. The projected change due to the impact of additional 0.5 °C and 1.5 °C warming comes with both opportunities and constraints for different crops across the AEZs of the region. Over the Sahel, it will likely become wetter with increasing greenhouse gas emissions (Nicholson, 2013; Sylla et al., 2013) resulting in the northward expansion of crops into the Sahel. There is also a projected increase in the length of the rainy season (LRS) over the Guinea and Savanna zones (Kumi & Abiodun, 2018). This may be responsible for the sustained levels of suitability despite the projected spatial contraction (decrease in suitability) under GWL1.5 and 2.0 but not at GWL 3.0 especially for the cassava and cereals where some areas become unsuitable due to the increased warming and the delay in month of planting for the different crop types in the zone. This provides opportunities for more cultivated land which may have a significant role in improving crop yield and production over the region and might influence the socio-economy of the region which is dependent on rainfed agriculture (Kurukulasuriya & Mendelsohn, 2006).

On the other hand, an additional 0.5 and 1.5 °C warming will also lead to constraint in suitability and spatial extent of some crops, most notably cassava, millet and plantain. In the context of GWLs, delta 0.5 & 1.5 °C may have limited impact on actual cropping with changes in suitability from highly suitable to suitable (e.g., millet and plantain). However, although not the purpose of this study, it will be very useful to quantify this change and investigate how it may influence crop yield and production over the regions, which are projected to decrease over the region in literature especially at GWL3.0 (Roudier et al., 2011; Challinor et al., 2014). Furthermore, the projected increase in risks to crop production as global warming levels rise from GWL1.5 to GWL2.0 and finally GWL3.0 is very evident. While all the crops are still suitable and can be cultivated over the region under GWL1.5 and to a level GWL2.0, the condition is much worse under GWL3.0 for all of the crops except cowpea and groundnut as some current suitable areas becomes unsuitable due to more warming, potentially compromising sustained crop production in West Africa. This further reiterates the importance and need for policymakers to ensure their commitment in meeting the Paris agreement or accords by member states of limiting global warming to 1.5 °C above pre-industrial level. This also calls for and put a responsibility on each member countries to implement their plan for addressing climate change challenge beyond 2020 aimed at limiting temperature below 2 °C (Rogelj et al., 2016), otherwise this may be devastating and further compound the woes of a highly vulnerable region like West Africa and with low adaptive capacity.

The impact of the projected global warming levels on PM varies for the different crop types; however, the influence was more pronounced on root and tuber and cereal crops especially cassava and maize respectively. In general, projected delay in PM from one to over four months may be experienced at over the region across the three AEZs under different warming levels. The projected delays to the four crops, cowpea, groundnut, pearl millet and plantain across the three warming levels and share common spatial characteristic pattern and sometimes in magnitude in the month of delay except for some pockets of area notably along the coastal areas or the Sahel zones where an early projected planting may expect for these crops. The impact of the projected delay in the planting and cultivation of these crops will be of concern to farmers (crop production and source of livelihoods) and policy makers (economic growth and international trade) which may further aggravate the impact of climate change on the regions. On the other hand, the impact of the global warming level on the PM is more drastic and obvious for cassava and maize. The projected change for cassava and maize show delays in PM are expected in the south, notably over the Guinea and central Savanna zone and early planting in the north in the Savanna-Sahel AEZs. The projected change in PM suggests an all-around planting season for these crops, which are very crucial and important to the inhabitants of the region in terms of livelihoods and economy in relation to crop production and food security as well as regional and international trade to boost the economy respectively especially cassava in which West African is one of the leading global producers of the crop.

Regional crop suitability, changes in planting months, adaptation and socio-economy in West Africa

Projected variability and shift in regional crop suitability and months of planting will be crucial to the socio-economic activity and regional trade in West Africa. Increased agricultural productivity can enhance economic growth resulting in industrial growth (Sultan & Gaetani, 2016). As seen from our findings above, projected crop suitability and notably the northern spatial expansion is one of the important factors that may enhance increased agricultural productivity. Increased suitability coupled with the planting during the best PMs, potentially linked to an increased Length of Rainy Season (LRS), may result in more cultivated land for crop growth and harvested areas. As earlier mentioned, the projected increased LRS with increased warming under RCPs 4.5 and 8.5 over the Guinea-Savanna and Sahel zones respectively (Kumi & Abiodun, 2018) and a wetter Sahel (Nicholson, 2013) can help improve agricultural productivity in the regions and have a positive impact on the economy and livelihoods of the inhabitants. Also, variation in suitability and planting months of the different crops can help increase the socio-economic livelihoods through regional trade amongst countries. Some countries with projected suitability expansion can improve their production through the availability of more cultivated land to meet their needs and create a market for countries with no or projected contraction in suitability to help offset their production deficits. Also, the variation in the period of planting for the different AEZs can create regional opportunity as crop production will be at different times of the years and this may help with regional and international trade amongst the different countries. The proposed change in planting month suggests a technical solution that can be used as adaptation strategy and this needs to be further explored. For example, Williams et al. (2018) have shown farmer’s understanding of changes in climate through their traditional knowledge by using traditional methods such as mulching, bush fallowing among others to address such challenge.

In regions where there are contractions in the spatial extent of suitability or reductions in the suitability index, improved adaptation strategies will be key to mitigate the impacts of these changes. With impact of GWL2.0 and GWL3.0 more drastic on crops such as cassava and maize, with their high socio-economic importance in West Africa, an improved understanding about the timing of adaptation cannot be overemphasized owing to the high vulnerability and low adaptive capacity of West Africa (Niang et al., 2014). This is important with the variation in projected changes in the month of planting for the different crop types. New knowledge about developing adaptation strategies, such as transformational adaptation as proposed by Rippke et al. (2016), may assist in mitigating the impact of GWL1.5, GWL2.0, GWL3.0 on crop suitability coupled with the projected changes in the month of planting over West Africa, notably for cereal and root and tuber crops, with spatial contraction and decrease suitability (Roudier et al., 2011; Challinor et al., 2014). These types of adaptation strategies could improve food security in the region through not only maximizing the yield potential of suitable areas, but also enhance regional trades amongst countries through trade-offs based on crop suitability status of each country (Rippke et al., 2016).

Summary and Conclusion

In this study, we assessed the impact of 1.5, 2 and 3 °C warming on crop suitability over West Africa. Climate characteristics result from 10 CMIP5 GCMs downscaled with RCA4 under RCP8.5 scenarios, were used as input into crop suitability model, Ecocrop for the past and future climate over West Africa. The impact of 1.5, 2, and 3 °C warming was computed using 1971–2000 as the reference period for six crops, millet, cassava, groundnut, cowpea, maize and plantain. Our findings are as follows:

• A low or no suitability to the north and high suitability to the south, separated marginal suitability line over West Africa in the historical climate for all the six crops, in general, marginal suitability lines are observed around 14°N for all the crops across the region except for plantain. Plantain has its marginal suitability line south of 12°N.

• At GWL1.5, there is a broadly similar spatial pattern of variation in suitability as the historical climate. However, a suitability shift (both spatial expansion and contraction simultaneously) is projected under 1.5 °C warming for cereals, legumes, groundnut along the central southern Sahel (around l13–14°N) and in the Guinea-Savanna zones for root and tuber, plantain.

• Projected changes in crop suitability and suitable areas under the GWL1.5 °C shows all the crops remain suitable across the three AEZs of West Africa, although with reduction in the SIV of some crops like cassava and the cereals.

• with GWL2.0 °C, the impact is more drastic on cereals and root and tubers with decrease in SIV of crops and a reduction in the suitability of some areas but are still suitable compared to legumes which have a relatively no change.

• The impact of GWL3.0 °C leads to a more devastating effect such as a high decrease in the crop SIV resulting in more suitable areas becoming less suitable and unsuitable for cultivation notably south of the region. In contrast, warming under GWL3.0 leads to a northern extension of suitable area in growing cereals and legumes in the central area of the southern Sahel. However, the suitable areas lost are far more than those gained with the increasing warming.

• the projected impact of GWL3.0 °C in comparison to GWL1.5 °C are most drastic on cereals and root and tuber crops with cassava the most impacted crop. The increase in warming, results in the loss of suitable areas in the southern Savanna and northern Sahel zone of the region become unsuitable for cassava in the south coast of Nigeria in the Guinea zone become marginally suitable for pearl millet and very high reduction, up to 0.3 in SIV for other crops. This further emphasizes the need for commitment to the Paris Accord by member country, and the benefit of limiting global warming to 1.5 °C that provides a suitable and favourable condition for cultivation and growth of the crops over West Africa.

• The projected changes in crop suitability under GWL2.0 °C are less than at GWL3.0 °C. The change shows that an additional 1.0 °C beyond GWL2.0 °C results in a decrease in SIV of the crop with drastic impact on the suitable area in the past climate leading to reduction in suitability of cultivated areas south of 14°N over West Africa This benefit in keeping global warming well below 2 °C compared to GWL3.0 °C cannot be overemphasized with the fast-growing population and food demand over West Africa.

• The impacts of the three GWLs for the planting month varies for the different crop types but is more pronounced on root and tuber and cereal crops especially cassava and maize respectively. In general, projected delay in PM from one to over 4 months may be experienced over the region across the three AEZs under the three GWLs for legumes, pearl millet and plantain.

• The projected change for cassava and maize show delays in PM are expected in the south notably over the Guinea and central Savanna zone and early planting in the north in the Savanna-Sahel AEZs.

• There is an increasing trend in the projected crop suitability change with increasing warming over the region and across the three warming for all the crops except cowpea and for groundnut between GWL1.5 and GWL2.0 °C.No change in trend value was observed for the planting for all the crops and across the three warming levels.

Although the present study has enhanced our understanding on the impact of GWL1.5, 2.0 and 3.0 °C warming on crop suitability and planting season over West Africa. Future studies may investigate the impact of GWL1.5, 2.0 and 3.0 °C warming on crop suitability over the region using more RCMs other than the single RCM with the different forcing GCMs used in this study as inputs into Ecocrop for more robust findings. Furthermore, the present study only considers six crops over the region, future work may use more crops classes such as horticultural (e.g., pineapple, tomatoes), fruit (e.g., oranges, mango), more cereals like wheat, rice; cash crops (e.g., oil palm), root and tuber (e.g., yam) to mention a few. Such research is needed to help guide policymakers at both the national and the regional level in reducing the impact and risk associated with food insecurity/scarcity in a changing climate in West Africa. Nevertheless, the present work has established that using RCM, RCA4 to downscale GCM simulations to drive a crop suitability model, can help improve our understanding on the impact of GWL1.5, 2.0 and 3.0 °C on crop suitability over West Africa. In addition, the study also shows the benefit of keeping global temperatures below 2 °C warming and most especially GWL3.0 °C on crop suitability growth and month of planting over West Africa.

Additional Information and Declarations

Competing Interests

Author Contributions

Data Availability

Jeff Price is an Academic Editor for PeerJ. The authors declare there are no competing interests.

Temitope Samuel Egbebiyi conceived and designed the experiments, performed the experiments, analyzed the data, prepared figures and/or tables, authored or reviewed drafts of the paper, and approved the final draft.

Olivier Crespo, Christopher Lennard, Jeff Price and Rachel Warren conceived and designed the experiments, performed the experiments, analyzed the data, authored or reviewed drafts of the paper, and approved the final draft.

Modathir Zaroug, Grigory Nikulin and Nicole Forstenhäusler conceived and designed the experiments, performed the experiments, authored or reviewed drafts of the paper, and approved the final draft.

Ian Harris conceived and designed the experiments, performed the experiments, prepared figures and/or tables, provided CRU historical dataset, and approved the final draft.

The following information was supplied regarding data availability:

The source code is available at:

http://cran.r-project.org/web/packages/dismo/index.html.

https://cran.r-project.org/web/packages/dismo/dismo.pdf.

The climate data is available at:

http://www.csag.uct.ac.za/cordex-africa/how-to-download-cordex-data-from-the-esgf/.

http://www.cordex.org/data-access/bias-adjusted-rcm-data/.

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
