# Peer review of "Investigating the potential impact of 1.5, 2 and 3 °C global warming levels on crop suitability and planting season over West Africa"

_PeerJ, doi:10.7717/peerj.8851_

## Round 0.1 · original submission · Major Revisions

Please carefully consider all the comments made by both reviewers. Detail all the changes that you make in the methodology (as suggested by Reviewer 2) and the modified text introduced in the body of the paper to address both reviewers' comments.

Reviewer 1 ·

Basic reporting

The study gives information about the possible impacts of three different warming levels in the agricultural sector of West Africa. This geographic area has already been detected as very vulnerable to climate change because most farmers in the region are rainfed farmers and the local economic activity strongly depends on agriculture. Consequently, the food security of the regions is expected to be compromised in the near future. The authors used climatic and crop suitability information to create suitability models under current and future climates. I think that the data and methods used are appropriate to fulfill their research goal and that findings are novel and interesting.
In general, in the text it is possible to see that there are several points missing and there are extra spaces or capital letters when they are not necessary. Authors need to read their manuscript again in order to find these mistakes. I have some other specific comments.

Experimental design

It is an original research with a well defined structure and question. The technical an ethical standards are rigorous and methods are being described with sufficient detail.

Validity of the findings

I think the research and results are novel and important. Data to replicate the model are provided. Conclusions are well stated and linkes to original research question.

Additional comments

Abstract
Even when it is possible to identify the most important components of an abstract, your results and discussion sections are confusing. Your main results and conclusions should be highlighted and clarity. There is also a lack of connectivity between sentences that need to be attended, especially in the last part of your abstract.
Line 24- The first sentence of your result section in the abstract is not clear. What do you mean with marginal suitability line?
Line 31- Is it of or from?
Lines 36- “are most dramatic (…)”- are more dramatic on cereals, roots and tuber crops?
Line 37- “All the projected changes in climatic area suitability…” Clarify that it is for simulated crop suitability that only considers climatic variables.

Introduction
This section gives a good introduction about the importance of the study. However, I think that the introduction could be enriched by giving more information about studies that have been working with climate change and agriculture in the same geographic area.

Line 46- There is an extra space.
Line 65- Which century? The 21st century? In that case, it would be more appropriate to use the most recent scenarios’ reference from the IPCC, which are the RCPs.
Line 65- A point is missing.
Lines 67- Have revealed that?
Line 98- ()
Lines 80-100- I think it would be better to summarize this information and not to write about it with so much detail. Instead, I would recommend mentioning some studies that have been dealing with topics related to this paper.
Line 101: Indicate which three are the different global warming levels and define IPCC in line 102.
Lines 111 and 116 –°C
112- A point is missing.

Data and Methods
Line 125- A point is missing
Line 129- Could you please give some references about the low adaptive capacity of the area and the reasons of why this adaptive capacity is so low in the agricultural sector.
Line 136- yam
Line 136- What does SSA mean?
Line 142- What do you mean by cash crops?
Line 153- What does RCMs mean? You have not defined it.
Line 158- EcoCrop? Ecocrop? Be consistent
Line 165- There is an extra point
Line 171- I am not sure that can call RCPs emission scenarios. You did not define an RCP. What are they? How many they are? What do they represent? Which are their unities?

Results
Line 268- Why do you write lat. Instead of latitude?
Line 309- February, March…
Line 403- “the region under?”




Discussion
It would be interesting to know if there is any field study in which it is possible to know if some adaptation strategies such as changing the planting months are already taken place in West Africa. Farmers’ traditional knowledge should be helping farmers realize that climate conditions are changing. How are they reacting to these changes? Is there a strong relationship between specific cultures and a cultivar? How difficult would it be for farmers to start growing what is climatically more suitable but not as attached to their culture?
It would also be interesting to mention the importance of other environmental factors to these plants’ suitability, such as soil. I understand that the focus of the work is climate change, but its is important to mention that there are other factors that might also be playing an important role in the future and that for now they are not being considered (limitation of the model).

Line 584- (
Line 593- ..
Line 597- under GWL3.0
Line 645- “These crops” “this crop”


Figure 1- There should be another way to better distinguish between the three areas. I think it is more important to distinguish between the three evaluated areas than showing the topography that is barely being mentioned in the text.

Annotated reviews are not available for download in order to protect the identity of reviewers who chose to remain anonymous.

Reviewer 2 ·

Basic reporting

Overall the reading is relatively easy thanks to the use of standard, scientific English with a relatively concise style. However many typographic or syntax errors spread throughout the text made me stop on certain sentences just to guess the meaning. In many of these errors a verb or a word is lacking (‘e.g lines 77 what are the projected impacts, ‘are’ is lacking; line 99 holding temperature increase below 1.5°C: "increase" is lacking, and see also lines 147, 224, 259, 273, 334, 344, 346, 356 to 359, 426..and then I stopped noting them).

The manuscript is well structured. Its introduction and background are mostly appropriate. However, the introduction could give less room to well-known generalities about climate change & Africa, and instead focus a little more on the different ways of evaluating the impact of climate change on agriculture, in order to better justify assessing ‘crop suitability’.

The particular concept of ‘crop suitability’, which is absolutely key to this study cannot be used throughout without any definition as if it were a standard concept (as is yield). There are many ways of envisaging if a crop is practicable somewhere, depending not only on biophysical variables but also on economic or social ones. So please give the definition you refer to in this study, either right in the introduction or in the ‘Data and Methods’ section.

The captions of figures 1-3 are not adequate. The variable displayed in column 1 (suitability) is not the same as in columns 2-4 (change in suitability), thus the title ‘Spatial distribution of crop suitability as simulated by Ecocrop over West Africa for Hist. (column 1) and (column 2-4)’ is misleading. It should be changed to something like ‘Spatial distribution of crop suitability as simulated by Ecocrop over West Africa for Hist. (column 1, left axis) and of change in crop suitability (right axis) at different global warming levels (GWL1.5, GWL2.0, GWL3.0) under RCP8.5 scenario (column 2-4)…’ Please apply the same principle to figures 4 and 5 (planting month and change in planting month).

Raw data are not supplied but this should not be an issue since the primary raw data used for the analysis and the software used are all available from the internet. The processed data of this paper has a great value precisely because the process is extremely demanding in terms of both expertise and labour time, so I don’t see why the team who obtained it would make it available without any counterpart.

Experimental design

The study is not particularly original, as it basically applies to a new case study the method from Ramirez-Villegas et al, 2013 (Ramirez-Villegas, J., Jarvis, A., Läderach, P., 2013. Empirical approaches for assessing impacts of climate change on agriculture: The EcoCrop model and a case study with grain sorghum. Agr. Forest Meteorol. 170, 67-78.). But the new case study is of substantial interest and novelty thanks to the assessment of the major crops of the region studied instead of only one. This gives the study the potential to better link the environmental analysis to social and economic issues, as it is conveniently said in the introduction.

The work is rigorous and well described except for the following key components:
- 1) An explicit and clear definition of the concept of crop suitability used in the paper is lacking

- 2) Crop suitability is many sentences confounded with change in crop suitability, resulting in a lot of confusing statements

- 3) Ecocrop model is not sufficiently detailed for a reader to fully understand the paper (and especially the differences in behaviour across the species accounted for in the study) without reading the references (Ramirez-Villegas et al, 2013, and Hijmans et al., 2001).

- 4) Default crop parameters are assumed to be adequate and the authors assume that they did not perform any calibration or new comparison between Ecocrop’s outputs and observed data, but nothing is said about how and by whom these default parameters were obtained or the model was previously evaluated against observed data. Lines 191-192 say ‘Previous studies have reported a good agreement between climate change impacts projections from Ecocrop model and other crop models’ and are followed by a number of citations, but the agreement referred to in this sentence has little to do with agreement between model and observed data. Who cares about a same thing said by many if these are as many liars ? Moreover, reading Ramirez-Villegas suggests that calibration is not straightforward and model results are extremely sensitive to these parameters. Therefore these key assumptions of the manuscript strongly jeopardize the way a reader can judge the validity of the study.

Validity of the findings

When crossing what is said in the paper with what is said in the key cited papers about Ecocrop model and its calibration, the validity of the finding of this manuscript is dubious.

Anyone with a little expertise on agriculture in West Africa would find at least one strong discrepancy between the real world and the suitability maps predicted by Ecocrop in this study for groundnut and cowpea under historical climate. The part of Senegal between 13.4° and 15.7° latitude and -16.6° and -14.3 ° longitude is known as the ‘groundnut basin’ and at least half of its the cultivated area has been holding groundnut for a century or so. And this zone was classified as unsuitable for groundnut in this study.

Due to this discrepancy, many key statements of the paper are abusive. The following one is especially irrelevant (lines 320-323): ‘These evaluation simulations demonstrate that the (RCA4-Ecocrop) captures the variation in suitability with different crops across the three AEZs of West Africa in the present-day climate and can serve as a baseline for evaluating the changes in crop suitability under global warming levels of 1.5 to 3oC over the region. The model also captures the growing season of crops over the region which varies with different months of the year’

As a result, the paper fails to link its environmental analysis with any socio-economic issues, which was the major condition for it to be a worthy contribution on top of that of previous works using Ecocrop under future climate projections.

Additional comments

Despite the manuscript has unacceptable weaknesses, it is obviously the result of great amount of dedicated work, and it does have a potential for becoming rigorous and valid enough to deserve publication in the Journal.

Therefore I would encourage the author to submit a carefully revised version addressing the weaknesses listed above. A proper recalibration of the Ecocrop model for all the species studied will be the most straightforward, albeit labour consuming solution to the issue #4 I raised in section ‘experimental design’.

The authors are of course free to find other ways to reconcile robustness of their conclusions with the uncertainties about the way simulated crop suitability matches observed crop distribution.

As a means to add value to the paper even when the uncertainties in the modelling exercise remain high, the authors should strengthen their discussion of the causes behind the differences between crops in responding to simulated climate change.

---

## Round 0.2 · Minor Revisions

I have carefully reviewed your responses to the comments made by reviewers and I am satisfied that you have satisfactorily addressed them.

However, before proceeding to accept the manuscript I would like you to carefully go through the text and correct a couple of typographical errors, and omissions, e.g. several sentences are missing articles, such a "the" and/or "a". Please upload the revised version as soon as it is convenient.

---

## Round 0.3 · accepted · Accept

The revised version includes the changes I had requested.